# DATA AUGMENTATION IN TRAINING CNNS: INJECTING NOISE TO IMAGES

## ABSTRACT

Noise injection is a fundamental tool for data augmentation, and yet there is no widely accepted procedure to incorporate it with learning frameworks. This study analyzes the effects of adding or applying different noise models of varying magnitudes to Convolutional Neural Network (CNN) architectures. Noise models that are distributed with different density functions are given common magnitude levels via Structural Similarity (SSIM) metric in order to create an appropriate ground for comparison. The basic results are conforming with the most of the common notions in machine learning, and also introduces some novel heuristics and recommendations on noise injection. The new approaches will provide better understanding on optimal learning procedures for image classification.

## 1 INTRODUCTION

Convolutional Neural Networks (CNNs) find an ever-growing field of application throughout image and sound processing tasks, since the success of AlexNet (Krizhevsky et al., 2012) in the 2012 ImageNet competition. Yet, training these networks still keeps the need of an "artistic" touch: even the most cited state-of-the-art studies employ wildly varying set of solvers, augmentation and regularization techniques (Domhan et al., 2015). In this study, one of the crucial data augmentation techniques, noise injection, will be thoroughly analysed to determine the correct way of application on image processing tasks.

Adding noise to the training data is not a procedure that is unique to the training of neural architectures: additive and multiplicative noise has long been used in signal processing for regression-based methods, in order to create more robust models (Saiz et al., 2005). The technique is also one of the oldest data augmentation methods employed in the training of feed forward networks, as analysed by Holmstrom & Koistinen (1992), yet it is also pointed out in the same study that while using additive Gaussian noise is helpful, the magnitude of the noise cannot be selected blindly, as a badly-chosen variance may actually harm the performance of the resulting network (see Gu & Rigazio (2014) and Hussein et al. (2017) for more examples).

The main reasons for noise injection to the training data can be listed as such in a non-excluding manner: first of all, injection of any noise type makes the model more robust against the occurrence of that particular noise over the input data (see Braun et al. (2016) and Saiz et al. (2005) for further reference), such as the cases of Gaussian additive noise in photographs, and Gaussian-Poisson noise on low-light charge coupled devices (Bovik, 2005). Furthermore, it is shown that the neural networks optimize on the noise magnitude they are trained on (Yin et al., 2015). Therefore, it is important to choose the correct type and level of the noise to augment the data during training.

Another reason for noise addition is to encourage the model to learn the various aspects of each class by occluding random features. Generally, stochastic regularization techniques embedded inside the neural network architectures are used for this purpose, such as Dropout layers, yet it is also possible to augment the input data for such purposes as in the example of "cutout" regularization proposed by Devries & Taylor (2017). The improvement of the generalization capacity of a network is highly correlated with its performance, which can be scored by the accuracy over a predetermined test set.

There has been similar studies conducted on the topic, with the example of Koziarski & Cyganek (2017) which focuses on the effects of noise injection on the training of deep networks and the possible denoising methods, yet they fail to provide a proper methodology to determine the level of

noise to be injected into the training data, and use PSNR as the comparison metric between different noise types which is highly impractical (see Section 3). To resolve these issues, this study focuses on the ways to determine which noise types to combine the training data with, and which levels, in addition to the validity of active noise injection techniques while experimenting on a larger set of noise models.

In the structure of this work, the effect of injecting different types of noises into images for varying CNN architectures is assessed based on their performance and noise robustness. Their interaction and relationship with each other are analyzed over (also noise-injected) validation sets. Finally as a follow-up study, proper ways on adding or applying noise to a CNN for image classification tasks are discussed.

## 2 DIFFERENT TYPES OF NOISE

Noise can be -somewhat broadly- defined as unwanted component of the image (Bovik, 2005). It can be sourced from the environment at which the image is taken, the device utilized to take the image, or the medium of communication that is used to convey the information of image from source to the receiver. According to its properties and nature, noise and image can be analytically decomposed as additive or multiplicative, but some of the noise types cannot be described by neither of these classes.

### 2.1 ADDITIVE NOISE

Let $f(\mathbf{x})$ denote an image signal. This signal can be decomposed into two components in an *additive* manner as $f(\mathbf{x}) = g(\mathbf{x}) + n(\mathbf{x})$ where $g(\mathbf{x})$ denoting the desired component of the image, and $n(\mathbf{x})$ standing for the unwanted noise component. Most commonly encountered variant of this noise class is Gaussian noise, whose multivariate probability density function can be written as:

$$f_n(\mathbf{x}) = \frac{1}{\sqrt{(2\pi)^n |\mathbf{\Sigma}|}} \exp\left(-\frac{1}{2}(\mathbf{x} - \mathbf{m})^T \mathbf{\Sigma}^{-1}(\mathbf{x} - \mathbf{m})\right) \tag{1}$$

where $\mathbf{m}$ and $\mathbf{\Sigma}$ denoting the n-dimensional mean vector and symmetric covariance matrix with rank n, respectively. In images, the mean vector $\mathbf{m}$ is generally zero, therefore the distribution is centered and the magnitude is controlled by the variance. This study also follows these assumptions.

### 2.2 MULTIPLICATIVE NOISE

Again, let $f(\mathbf{x})$ denote an image signal. This signal can also be decomposed into respective desired and noise components as $f(\mathbf{x}) = g(\mathbf{x})(1 + n(\mathbf{x}))$. The noise component in this model is called *multiplicative* noise. The most common variant in this case is called *speckle* noise, which may have different density functions, and in this study Gaussian is assumed. Similar with the additive noise, the mean is assumed to be 0 and the magnitude refers to the variance.

Speckle noise can be encountered in coherent light imaging such as in the cases of SAR images (Ding et al., 2016) and images with laser-based illumination, but they may also be observed in other digital images (Bovik, 2005).

### 2.3 OTHER TYPES OF NOISE

There exists many other noise instances that cannot be modeled by additive or multiplicative decompositions. Most common of these types are listed below, whose effects on the performances of CNNs are also analysed in this study.

**Salt and pepper (S&P) noise.** This noise manifests itself as a basic image degradation, for which only a few pixels in an image are noisy, but they are extremely noisy in a way that pixels are either completely black or white. The magnitude of this noise is the probability of a pixel to be completely black (i.e. pepper), completely white (i.e. salt), or stay unchanged. The probabilities for pepper and

salt cases are assumed to be equal, and total probability of the degradation of a pixel is referred as the the magnitude of the noise.

**Poisson noise.** Also referred as *photon counting* noise or *shot* noise, this noise type has a particular probability density function:

$$f_n(\mathbf{k}) = \frac{e^{-\lambda}\lambda^{\mathbf{k}}}{\mathbf{k}!} \tag{2}$$

where $\lambda$ stands for both the variance and the mean of the distribution. As Poisson noise is signal-dependant, it does not have a direct magnitude parameter similar to other noise types, therefore a magnitude factor $c$ is used that divides the intensity values of all pixels from which the distribution is sampled, and returned to the original range by multiplying to the same factor.

**Occlusion noise.** Although it is not generally referred as a noise type, occlusion of important features can happen for a number of reasons in an image: image may be cropped, damaged or a particular object in it may be hidden by an obstacle such as a tree. This noise is realized with zero-intensity squares appearing on the image, and the magnitude is determined according to the size of the square as the shape of the occluding object does not have a large impact on the final performance of the model (Devries & Taylor, 2017).

As listed in this section, five different types of noise and their various combinations are added or applied to the images, with varying magnitudes. Robustness against each of these noise types is also assessed.

## 3 CHOOSING THE RIGHT METRIC: PSNR VS SSIM

Different types of noise are easy to compare when they are sampled from the same distribution, such as in the case of additive Gaussian noise and speckle noise. However, it is sometimes impossible to assess two different metrics in the same context because of varying magnitude parameters and sensitivity of the model to each of these types. In this case, it becomes imperative to use an intermediary metric that will ease the comparison process for noise types.

In general, Peak Signal-to-Noise Ratio (PSNR), and similarly Mean Squared Error (MSE) are the most commonly used quality metrics in the image processing field. For an 8-bit two-dimensional MxN image $\hat{f}(n_1, n_2)$ and its noise-free counterpart $f(n_1, n_2)$, the MSE is defined as

$$MSE = \frac{1}{MN} \sum_{n_1=0}^{M-1} \sum_{n_2=0}^{N-1} [f(n_1, n_2) - \hat{f}(n_1, n_2)]^2. \tag{3}$$

From above definition, PSNR (in dB) can be derived.

$$PSNR = 10\log_{10}\left\{\frac{255^2}{MSE}\right\} \tag{4}$$

There are several limitations of using PSNR as the image quality metric of a data set: it is shown that PSNR loses its validity as a quality metric when the content and/or codec of the images are different as in that case the correlation between subjective quality and PSNR is highly reduced (Huynh-Thu & Ghanbari, 2008). Also, even though the sensitivity of PSNR to Gaussian noise is very high, the metric is unable to present similar performance for different types of perturbation (Avcibas et al., 2002) (Horé & Ziou, 2010).

There exists another widely accepted image quality metric called Structural Similarity (SSIM), which resolves or alleviates some of the above-listed problems. The Structural Similarity between two non-negative image signals **x** and **y**, whose means, standard deviations and covariance are denoted by $\mu_x$, $\mu_y$, $\sigma_x$, $\sigma_y$ and $\sigma_{xy}$ respectively, can be expressed as:

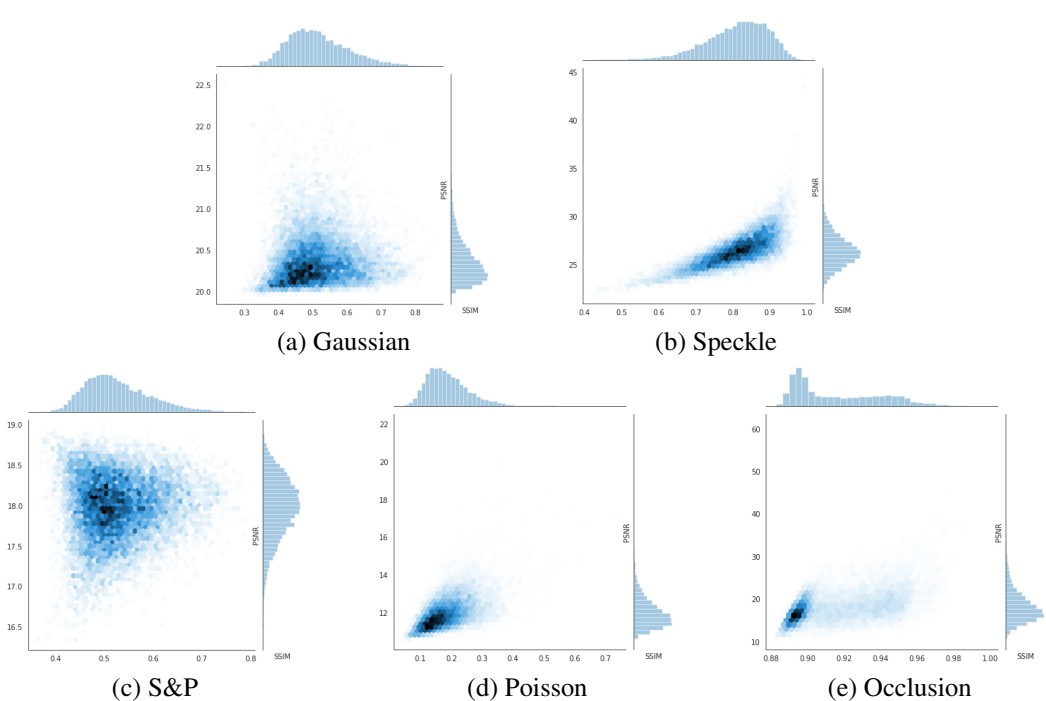

Figure 1: "hexbin" plots of the PSNR and SSIM values for different noise types over *Imagewoof* dataset, a subset of ImageNet with training size of 13000 samples.

$$SSIM = \frac{(2\mu_x\mu_y + C_1)(2\sigma_{xy} + C_2)}{(\mu_x^2 + \mu_y^2 + C_1)(\sigma_x^2 + \sigma_y^2 + C_2)} \tag{5}$$

where $C_1$ and $C_2$ are constants to avoid instability (Zhou Wang et al., 2004). This metric combines luminance, contrast and structure information in order to provide an assessment of similarity in the range from 0 to 1, where 1 stands for highest similarity.

In image classification tasks, the objective is mostly to classify the depicted objects or beings according to the human perception. Therefore, it is crucial for the used metric to be consistent with human opinions: it is shown in several studies that SSIM provides a quality metric closer to average opinion of public than PSNR (Kim et al., 2019) (Zhou Wang et al., 2004).

Furthermore, when sampled from the same noise distributions over identical images of a data set, distribution of PSNR values of the noisy images have significantly high kurtosis than their SSIM counterparts in every noise type except S&P. This behavior is demonstrated over a subset of ImageNet dataset (Deng et al., 2009) called *Imagewoof* (can be accessed via github.com/fastai/imagenette), with 10 classes and 12454 total number of images on the training set. Each type of noise listed in Section 2 is applied to each image, sampling from the same distribution, and quality metrics are recorded. The utilized noise magnitudes are 0.2 variance for Gaussian noise, 0.2 variance for speckle noise, 0.05 total probability for S&P noise, 0.2 scaling factor for Poisson noise, and 0.3 relative side length for occlusion noise; which are all chosen to provide sensible values from the metrics. The distribution of the metrics can be seen at Figure 1 over the axes

| Gaussian | | Speckle | | S&P | | Poisson | | Occlusion | |
|---|---|---|---|---|---|---|---|---|---|
| PSNR | SSIM | PSNR | SSIM | PSNR | SSIM | PSNR | SSIM | PSNR | SSIM |
| 4.263 | **0.074** | 4.426 | **0.644** | **0.253** | 0.278 | 6.221 | **2.030** | 7.286 | **-0.830** |

Table 1: Kurtosis values of quality metric distributions over different types of noise

of each subfigure, and the kurtosis values are noted in Table 1. This is interpreted as PSNR having propensity to produce more outliers than SSIM for the same levels of noise (Westfall, 2014).

For the reasons listed above, SSIM will be used as the primary metric throughout this study.

# 4    EXPERIMENTS AND RESULTS

Effects of injecting different noise types into training data are evaluated for different magnitudes and types of the noise as listed in Section 2. The chosen datasets are two different subsets of ImageNet dataset, namely *Imagenette* and *Imagewoof*, each consisting of 10 different classes with 12894 and 12454 training samples respectively and 500 test samples (both can be accessed via github.com/fastai/imagenette). Former dataset contains ten easily classified classes of the original set, while the latter task is the classification of ten different dog breeds and require the network to successfully learn the particularly small features of the classes. Image data range is henceforth from 0 to 1.

In order to select the magnitudes of each noise component, a sweep for mean SSIM (MSSIM) of an array of noise magnitudes for each noise type is conducted over 200 images of Imagewoof dataset. Resulting graph can be seen in Figure 2. Very similar results are also observed when the same procedure is conducted on Imagenette dataset. According to the shapes of the curves, a quadratic polynomial is fitted to the relative side length of occlusion noise, and logarithmic polynomials are fitted to the rest. Look-up table for the fittings can be seen at Table 2, which are also the degradations applied in the experiments. Exemplary application of the noise can be seen at Figure 3.

As the training model, a 18-layer deep residual network (ResNet18V2) as proposed by He et al. (2016) is chosen, because of the fact that it is a well-known architecture and also sufficiently deep: Bengio et al. (2011) demonstrate that performance of the deep networks are more sensitive to data augmentation than their more shallow counterparts. The residual connections and layering structure of ResNet18V2 exhibit similar architectural properties of the most often utilized CNNs in the field.

Adam solver with learning rate 1e-4 is preferred for training. Models are trained for 20 epochs. No dropout or weight decay is used for regularization purposes, and Batch Normalization layers are used in accordance with He et al. (2016).

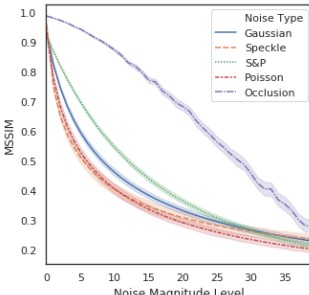

Figure 2: MSSIM vs Noise

| | Noise Types | | | | |
| --- | --- | --- | --- | --- | --- |
| MSSIM | Gaussian var. | Speckle var. | S&P prob. | Poisson magn. | Occlus. length |
| 0.25 | 0.0341 | 0.3355 | 0.1288 | 0.1046 | 1.1908 |
| 0.5 | 0.0085 | 0.0515 | 0.0461 | 0.0222 | 0.9078 |
| 0.7 | 0.0028 | 0.0115 | 0.0203 | 0.0064 | 0.6308 |
| 0.8 | 0.0016 | 0.0054 | 0.0134 | 0.0035 | 0.4753 |
| 0.9 | 0.0009 | 0.0026 | 0.0089 | 0.0019 | 0.3086 |

Table 2: Look-up table for the noise levels and MSSIM values

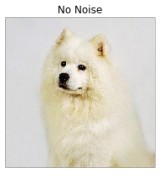
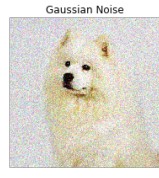
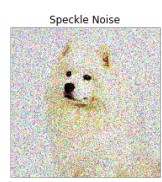
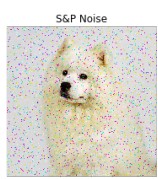
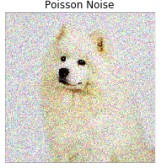
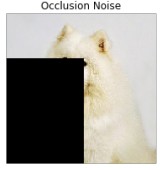

Figure 3: Exemplary noise applications for MSSIM = 0.5 from Table 2

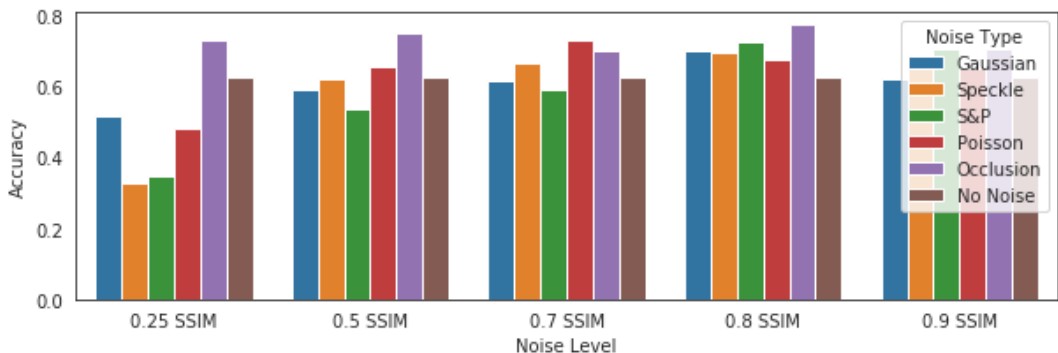

Figure 4: Accuracy of models over clean test set for Imagenette dataset.

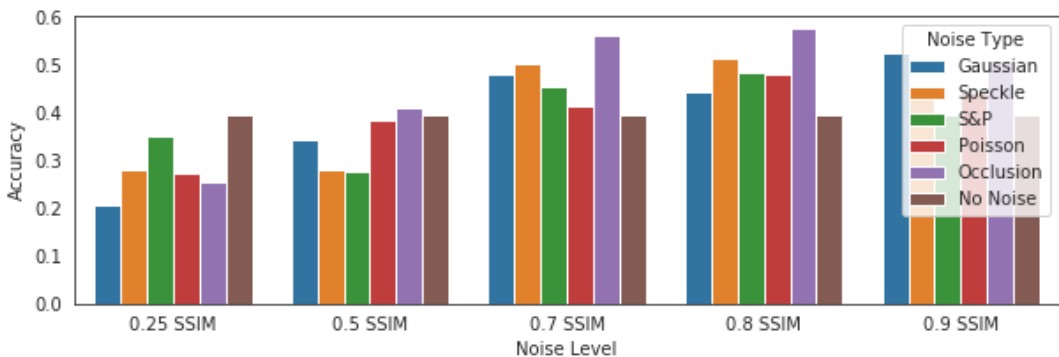

Figure 5: Accuracy of models over clean test set for Imagewoof dataset.

### 4.1 COMPARISON OF NOISY AND VANILLA TRAINING

Chosen CNN architecture is trained with noise injected to the training data for all noise models described in Section 2, for magnitudes corresponding to respective MSSIM values from Table 2. Total number of 52 networks are trained (25 on noisy data and 1 on original data for each dataset). The categorical accuracy of each trained network on the validation set can be seen at Figures 4 and 5, and the accuracies of the CNNs depending on the noise they are trained with for noisy test set can be observed at Figure 8. The latter results can be considered as the robustness test of the trained networks. One of the most important features of these heatmaps, robustness of the models against the noise injected to their input, is also plotted for each dataset individually in Figures 6 and 7.

## 5 DISCUSSION

There are several confirmations to acquire from this set of results for the literature: first of all, there exists a trade-off between noise robustness and clean set accuracy. Yet contrary to the common notion, we believe that the data presents a highly valid optimum for this exchange in our study. As it can be seen from Figures 6 and 7; in order to create a robust model against particular kind of noise while maintaining the performance of the model, one must apply a level of degradation that results in 0.8 MSSIM over training data. We believe that as long as the noise or perturbation is somewhat homogeneously distributed, this rule of thumb will hold for all image classification tasks. However, the same thing cannot be said for non-homogeneously distributed noise models, as SSIM (and also PSNR as demonstrated in Section 3) fails to capture the level of degradation appropriately for such a verdict (see Occlusion results in Figures 6 and 7).

A second confirmation of the current literature is the fact that the neural networks optimize on the noise level they are trained with, as seen again at Figures 6 and 7, and also the diagonals of Figure 8. Yet, the level of this optimization is quite small after 0.5 MSSIM, featuring similar robustness for each trained model. Therefore, it is not particularly necessary to determine the noise level of

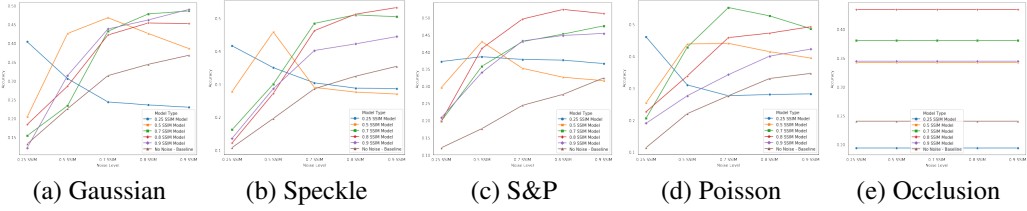

Figure 6: Accuracy (robustness) plots of the models to their noise type, Imagewoof dataset.

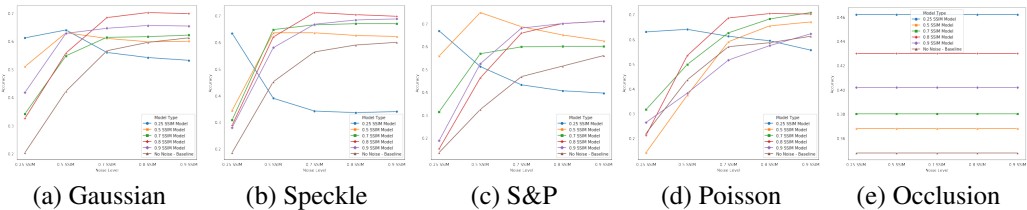

Figure 7: Accuracy (robustness) plots of the models to their noise type, Imagenette dataset.

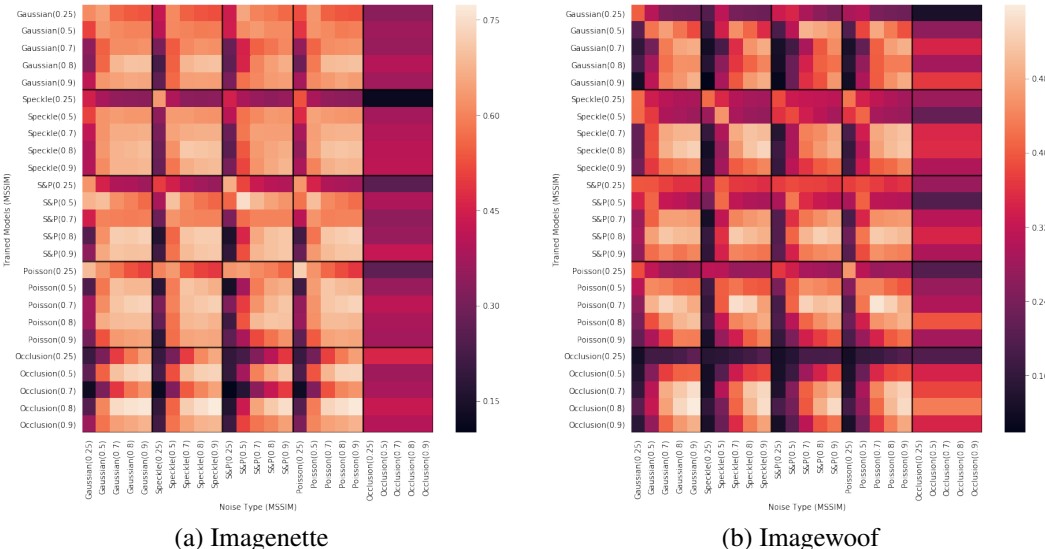

Figure 8: Accuracy (Robustness) heatmaps of each model to test sets augmented with different noise types. High accuracy on the diagonals are signs of noise level optimization of neural networks.

a dataset, or sample the noise from a predetermined interval, as long as the MSSIM does not drop below 0.5, in which case noise removal techniques need to be considered for better models.

As noted above, occlusion noise type will not be thoroughly analyzed in this section because of the fact that the quality metric has failed to provide sufficient comparative data for this discussion. Yet, the performance data and the lack of robustness the other models exhibit towards this particular noise type shows that "cutout" regularization as presented by Devries & Taylor (2017) is a crucial part of data augmentation in addition to any other perturbation or noise injection technique. A way to further extend the contribution of this method would be to alternate the intensity level of the patches from 0 to 255 for 8-bit images, which can be a topic of another research.

For the rest of the noise types; Gaussian, speckle and Poisson noises are observed to increase the performance of the model while boosting the robustness, and their effects exhibit the possibility of interchangeable usage. For image classification tasks involving RGB images of daily objects, injection of only one of these noise types with above-mentioned level is believed to be sufficient as repetition of the clusters can be observed in Figure 8. Among these three, Gaussian noise is recom-

mended considering the results of model performance. S&P noise contamination, on the other hand, may not be resolved by injection of the former noise types as the other models are not sufficiently robust against it. Therefore, at this point one of the two methodologies are suggested: either S&P noise can be removed by simple filtering techniques, or S&P noise can be applied in an alternating manner with Gaussian noise during data augmentation. Former approach is recommended for the simplicity of the training procedure.

The constant behaviour of the models towards occlusion noise in Figures 6, 7 and 8 unfortunately does not have a satisfactory explanation, despite several diagnostics of the training procedure. A longer training procedure, which was not feasible in our experiment because of the model count, may resolve these undesirable results.

## 6 CONCLUSION

In this study, an extensive analysis of noise injection to training data has conducted. The results confirmed some of the notions in the literature, while also providing new rule of thumbs for CNN training. As further targets of research, extension of "cutout" regularization as described in the above paragraphs, and the distribution behavior of the SSIM and PSNR metrics in Figure 2 with regards to the work of Horé & Ziou (2010) may be pursued.

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
