# OpenReview forum: "Data Augmentation in Training CNNs: Injecting Noise to Images"
_ICLR.cc/2020/Conference — Reject_

### Official Review · AnonReviewer3 · 2019-10-20
**Official Blind Review #3**

**Rating:** 3

**Review:**

The paper studies the effect of various data augmentation methods on image classification tasks. The Authors propose the Structural Similarity (SSIM) as a measure of the magnitude of the various types of data augmentation noise they consider. The Authors argue that SSIM is superior to PSNR as a measure of the intensity of the noise, across various noise types.

The idea of using SSIM as a unified measure for noise in the context data augmentation in images is novel AFAIK and is neat IMO, because as the authors point, SSIM provides a more perceptually-driven distance measure between images than RMS or PSNR. One of the results of the paper is that a SSIM value of 0.8 is a good rule of thumb for choosing the magnitude of the noise irrespectively to the type of noise. This is a useful and interesting result.

Nevertheless, at this stage I am inclined to reject the paper, because I feel that the main claim is not sufficiently substantiated. The argument that SSIM provides a more universal (less noise-type-dependent) measure of strength than, say, PSNR in the context of data augmentation is not substantiated in the experiments. While intuitively the claim makes sense to me, it is hard to draw this conclusion when SSIM is not compared to any other metric (RMS, PSNR) as measure of strength for data augmentation.

Why is a larger Kurtosis detrimental for measuring the strength of data argumentation? Is there any evidence that links low Kurtosis to a better measure data augmentation strength? This is another question that could be addressed if SSIM were compared to other data augmentation strength metrics.

IMHO the paper could have been made much stronger if it had the analog of Figure 5 for other measures of the noise (e. g. PSNR, RMS). If the results showed that SSIM is superior to them, I would learn a useful and insightful lesson form the paper. In its current form, I feel that the Authors made the first step in a very interesting direction but did not go far enough to substantiate their claims.

**Experience Assessment:**

I have published one or two papers in this area.

**Review Assessment: Checking Correctness Of Derivations And Theory:**

I assessed the sensibility of the derivations and theory.

**Review Assessment: Checking Correctness Of Experiments:**

I assessed the sensibility of the experiments.

**Review Assessment: Thoroughness In Paper Reading:**

I read the paper at least twice and used my best judgement in assessing the paper.

---

### Official Review · AnonReviewer1 · 2019-10-23
**Official Blind Review #1**

**Rating:** 1

**Review:**

This paper studies the different types of noises that could be added to the training image dataset while training an CNN model for classification. They study 5 different types of noise functions: Gaussian, Speckle, Salt and Pepper, Poisson, Occlusion.

Pros:
1. Rigorously studying how to augment training data for CNN is important.


Cons:
1. The primary question is novelty . - what is the research contribution of this dataset? They are running experiments of 5 different known noise functions, on two image datasets, on a single deep learning models. There are no fundamental research questions or hypothesis. This is a mere running of few experiments - known methods and known approaches.

2. Are the results generalizable? Answer is no! The results on shown on two subsamples of Imagenet datasets for only ResNet 18 model. Maybe for this combination speckle noise (and not Gaussian, as pointed out in the comments by the authors) is better. How can the assure that for a different dataset, model, task combination the same speckle noise would perform better ?

3. Improvement suggestion: What I would ideally look in this topic, is a method to automatically study the properties of the training data images (study the distribution) and conditional on this distribution recommend the best noise type and noise intensity. Thus, the whole story of noise injection could be made dynamic for a dataset, model, task combination

4. Writing of the paper could be improved: 1. The need for noise based augmentation of is well known (Section 1). 2. The different kinds of noise functions are mostly text book knowledge (Section 2). 3. The different image quality metrics written here - MSE, PSNR, SSIM are also text book knowledge (Section 3). Overall, the first 4 pages of the paper are redundant and could be compressed into 1 page. Would like to read more on the experiments, analysis, and maybe automation of noise selection techniques in different kinds of tasks - segmentation, text classification, seq2seq etc.

5. Choose very naive noise (or augmentation) functions: In general, the approach of augmentation of training data has evolved so much in the literature, that adding noise is hardly in practice.
1. "The Effectiveness of Data Augmentation in Image Classification using Deep Learning" - Style Transformation
2. "Improving Deep Learning using Generic Data Augmentation" - Geometric and Affine Transformation
Thus, the study on data augmentation should be performed across all these different transformation functions on training data and using only noise function is naive and is incomplete.

**Experience Assessment:**

I have published one or two papers in this area.

**Review Assessment: Checking Correctness Of Derivations And Theory:**

I carefully checked the derivations and theory.

**Review Assessment: Checking Correctness Of Experiments:**

I carefully checked the experiments.

**Review Assessment: Thoroughness In Paper Reading:**

I read the paper thoroughly.

---

### Official Review · AnonReviewer2 · 2019-10-24
**Official Blind Review #2**

**Rating:** 3

**Review:**

This paper aims at analyzing the effect of injecting noise to images as data augmentation in training CNN for the image classification task. Based on the SSIM metric (which is shown to be a better metric than PSNR), different noise level on a set of different kinds of noise are explored. Experimental results on two sub-datasets of ImageNet suggest that Speckle noise would lead to better CNN models.

Even though the simulations appear seemingly convincing, and the conclusion is somewhat interesting to me: speckle noise is recommended which contradicts the general usage of Gaussian noise. The results is too specific to both the model chosen resnet18v2 and also in the chosen dataset. Besides, my bigger concern is that the contribution of this work is highly limited, since there are a bunch of data augmentation techniques: cropping, flipping, color space transformation, rotation, noise injection, etc. Given this broad selection of data augmentation, as far as I know, noise injection is not the most effective nor popular one.  In fact, random cropping is the mostly used one that established past a few benchmark CNN models in the imageNet classification task, e.g, ResNet, DenseNet, etc. As such, it would be more convincing if it can be shown that proper noise injection can boost the recognition performance on the ImageNet task.

Minors:
Abstract line 8, and also introduces -> and also introduce

**Experience Assessment:**

I have read many papers in this area.

**Review Assessment: Checking Correctness Of Derivations And Theory:**

I assessed the sensibility of the derivations and theory.

**Review Assessment: Checking Correctness Of Experiments:**

I assessed the sensibility of the experiments.

**Review Assessment: Thoroughness In Paper Reading:**

I read the paper at least twice and used my best judgement in assessing the paper.

---

### Author Response · Authors · 2019-09-27
**Change of code address**

Due to the doubts of anonymity with regards to Google Drive shares, the link in the submission is deactivated. Code can be accessed via this new anonymous link: https://gofile.io/?c=GeXNVQ

Sorry for the inconvenience.

---

### Author Response · Authors · 2019-10-06
**Correction of a crucial typo and a small additional study**

Dear reviewers and readers,

As edit option for our paper still appears to be disabled (any recommendations regarding to this issue is highly appreciated), we are publishing an important correction and a small additional study via comment option.

The correction is due to a typo in the fourth paragraph of the 5th section (Discussion), that starts with the words "For the rest of the noise types...": Our final recommendation for the type of noise to be injected among Gaussian, speckle and Poisson noise models is NOT Gaussian, but speckle. The reasoning can be visually seen especially in the Figure 8(b), where speckle noise provides considerably better robustness than Gaussian noise. Furthermore, in Figures 4 and 5 neural models trained with speckle noise performs better than their Gaussian counterparts on six occasions, while the contrary is true only for three occasions (one case is nearly equal). This can be explained by speckle noise being feature-selective, targeting the high-intensity regions more than the low-intensity ones and thus allowing the neural network to better generalize for minor features. Although these observations have been made before the completion of the study, we are sorry for the crucial typo that will also be corrected in the original paper as soon as he editing is enabled. If reviewers approve, it is also possible to add this short explanation.

Small additional study is made upon the recommendation of a colleague, regarding to a conclusion reached by Koziarski & Cyganek (2017) that "noise as a form of regularization on top of other regularization techniques, namely weight decay and dropout, does not improve the classification accuracy". In the mentioned study, the properties of the noise and other regularization techniques applied to reach this conclusion are not disclosed, therefore we have made a series of experiments to determine the robustness and accuracy of the dropout-regularized CNN models with and without the noise injection procedure as advised in our study.  Initial reasoning behind not using such techniques was to comply with the ResNetV2 architecture of He et al. (2016).

We have chosen an aggresive dropout level of 0.5, and applied one of the three noise models; namely speckle, s&p and occlusion noise, at 0.8 MSSIM (see Table 2 from the study). For both datasets, we run the experiments for four different cases (w/o and w/noise, w/o and w/dropout), and trained each model for five times. The resulting loss metrics for each epoch is aggregated by the minimum value among five trials. The test set is composed of noise-free images of original datasets and their slightly noisy (0.9 MSSIM) counterparts for each noise type, in order to test for the model robustness with accuracy at the same time. The plots for each dataset can be seen in the following anonymous link: https://imgur.com/a/REothM6

The results are conforming with our discussions in the study, therefore there is no need for any change in the main structure, and this additional work can be seen as a confirmation of our findings. Noise injection, when appropriately constructed, can also be a good regularization technique especially with relatively difficult datasets (observe that the effect of noise injection is much greater in Imagewoof dataset which is substantially more challenging than Imagenette). The code for this additional work can be reached via: https://gofile.io/?c=2nXLAV

As we understand the policy of ICLR regarding to the follow-up studies, these results will be added to the paper depending on the decision of the reviewers.

References

Kaiming He,  Xiangyu Zhang, Shaoqing Ren, and Jian Sun. Identity mappings in deep residual networks. In Bastian Leibe, Jiri Matas, Nicu Sebe, and Max Welling (eds.), Computer Vision– ECCV 2016, pp. 630–645, Cham, 2016. Springer International Publishing.  ISBN 978-3-319-46493-0.

Micha Koziarski and Boguslaw Cyganek. Image recognition with deep neural networks in presence of noise: dealing with and taking advantage of distortions. Integrated Computer-Aided Engineering, 24:1–13, 08 2017. doi: 10.3233/ICA-170551.

---

### Decision · Program_Chairs · 2019-12-19

**Decision:**

Reject

**Comment:**

This paper studies the effect of various data augmentation methods on image classification tasks. The authors propose the structural similarity as a measure of the magnitude of the various types of data augmentation noise they consider and argue that it is outperforms PSNR as a measure of the intensity of the noise. The authors performed an empirical analysis showing that speckle noise leads to improved CNN models on two subsets of ImageNet. While there is merit in thoroughly analysing data augmentation schemes for training CNNs, the reviewers argued that the main claims of the work were not substantiated and the raised issues were not addressed in the rebuttal. I will hence recommend rejection of this paper.